# Wear Characteristics of Dental Ceramic CAD/CAM Materials Opposing Various Dental Composite Resins

**DOI:** 10.3390/ma12111839

**Published:** 2019-06-06

**Authors:** Bora Gwon, Eun-Bin Bae, Jin-Ju Lee, Won-Tak Cho, Hyun-Young Bae, Jae-Won Choi, Jung-Bo Huh

**Affiliations:** Department of Prosthodontics, Dental Research Institute, Institute of Translational Dental Sciences, BK21 PLUS Project, School of Dentistry, Pusan National University, Yangsan 50612, Korea; vvvora722@gmail.com (B.G.); 0228dmqls@hanmail.net (E.-B.B.); ljju1112@hanmail.net (J.-J.L.); joonetak@hanmail.net (W.-T.C.); h.02@daum.net (H.-Y.B.)

**Keywords:** monolithic zirconia, lithium disilicate, leucite, composite resin, wear

## Abstract

The aim of this study was to evaluate the wear properties of opposed dental ceramic restorative CAD/CAM materials and several posterior direct restorative composite resins. Three kinds of dental ceramics CAD/CAM materials (monolithic zirconia, lithium disilicate, leucite) and four dental composite resins—that is, MI Gracefil, Gradia Direct P, Estelite Σ Quick, and Filtek Supreme Ultra—were used in this study. For each of the 12 groups (three ceramics × four composite resins), five each of a canine-shaped ceramic specimen and a cuboidal shape opposing composite resin were prepared. All of the specimens were tested in a thermomechanical loading machine (50 N, 100,000 cycles, 5/55 °C). Wear losses of ceramic specimens and composite resin specimens were evaluated using a three-dimensional profiling system and an electronic scale, respectively. Statistical analyses were performed using the Kruskal–Wallis test and Mann–Whitney U test with Bonferroni’s correction. Zirconia showed significantly less volumetric loss than lithium disilicate or leucite regardless of composite resin type (*p* > 0.05/3 = 0.017), and that Estelite Σ Quick showed significantly more weight loss than Filtek Supreme Ultra, MI Gracefil, or Gradia Direct P regardless of ceramic type (*p* > 0.05/6 = 0.083). Zirconia showed less volumetric loss than lithium disilicate or leucite. Some composite resins opposing ceramics showed considerable weight loss.

## 1. Introduction

As a result of increasing interest in esthetics, various tooth-colored restorative materials are being widely used for tooth restoration, and in particular, dental ceramics and composite resins are being increasingly used as restorative materials for posterior teeth [1,2]. Dental ceramics have excellent biocompatibility, esthetic, low plaque accumulation, low thermal conductivity, and high color stability properties [3,4]. On the other hand, all-ceramic materials like feldspathic-, glass-, and glass-reinforced ceramics have been considered as restorative materials only for single crown restorations for some time because they are brittle and have low tensile strengths [1,3,5,6]. However, the developments of computer-aided-design/computer-aided-manufacturing (CAD/CAM) technologies and of reinforced dental porcelains—such as aluminum oxide, leucite, lithium disilicate, and zirconium oxide—have produced esthetically satisfying results without porcelain veneering as compared with traditional dental ceramics. In addition, these materials excellent physical properties and thus, are widely used in fixed prostheses [7,8,9]. However, the disadvantage of dental ceramics is that they can cause significant enamel wear on opposing teeth [10].

Composite resins are the most widely used dental restorative materials, and provide satisfactory esthetics and allow teeth restoration with minimal dental invasion [11]. However, traditional composite resins are only recommended for limited areas due to their relatively poor mechanical properties and the risk of restoration failure due to breakage and low wear resistance [12,13]. As a result, manufacturers have been developing more rigid and fracture-resistant composite resins [14,15] and the posterior composite resins currently available contain fillers of various hardnesses and sizes that enable them to withstand the masticatory forces generated by posterior teeth [16,17].

Various types of restorations in the oral cavity have occlusal contact with each other, and continuous chewing will inevitably wear restorations. Many studies have been conducted on the wear characteristics of various restorative materials—amalgam [18], gold [19,20], dental composite resin [10,15,19], zirconia [1,10,21,22], porcelain [1], and ceramic [10,22,23]. An ideal restorative material should have wear properties similar to those of enamel, such that wear proceeds in both at similar rates [24,25]. Silva et al. [22] found that lithium disilicate glass ceramics have higher wear resistance and cause less wear on opposing enamel than feldspathic ceramics used as veneers. Jung et al. [1] showed opposing natural tooth abrasion caused by zirconia is less than that caused by feldspathic dental porcelain and the process for polishing the surface of zirconia can further reduce the wear of opposing enamel; interestingly, zirconia with glazing is more abrasive than polished zirconia [1,21,24].

Commercially available composite resins can be classified as hybrid, microfilled, microhybrid, or nanohybrid resins [16], and the functions and performances of these resins are influenced by the size and makeup of filler particles, resin composition, degree of polymerization, and the bond between filler and the resin matrix [26,27,28,29]. Nanohybrid composites, which are hybrid composite resins containing prepolymerized nanofillers, have excellent mechanical properties and are widely used for anterior and posterior restorations [30,31]. Recently developed composite resin materials have greatly improved resistance to wear attributed to filler content and resin matrix quality improvements [10]. Fine particle size fillers increase wear resistance by reducing inter-particle distance [32], and microfilled composite resins containing colloidal silica fillers cause little enamel wear as compared with composite resins containing various kinds of fillers [33].

Occlusal contact in the oral cavity occurs between a variety of combinations of restorations, and the types of restorative materials involved affect wear properties [19]. Previous studies on this topic have mainly focused on the wear characteristics of restoration material/enamel systems [18], and relatively few studies have examined the wear characteristics of restoration material/restoration material systems [19]. Accordingly, the purpose of this study was to evaluate the wear properties of dental ceramic restorations opposing various posterior direct restorative composite resins. The null hypothesis tested for this study was that no significant difference could be detected in the wear properties among the materials under this study.

## 2. Materials and Methods

Monolithic zirconia (Zirkonzahn prettau, Zirkonzahn GmbH, Bruneck, Italy), lithium disilicate glass ceramic (Rosetta SM, HASS, Kangneng, Korea), and leucite glass ceramic (Rosetta SM, HASS, Kangneng, Korea) were fabricated into 60 ceramic specimens. For dental composite resin specimens, 15 of 4 different types of commercially available direct composite resins were subjected to antagonistic wear testing; that is, one nanohybrid composite (MI Gracefil, GC, Tokyo, Japan), two microhybrid composites (Gradia Direct P, GC, Tokyo, Japan; Estelite Σ Quick, Tokuyama Dental, Tokyo, Japan), and one nanofilled composite (Filtek Supreme Ultra, 3M ESPE, St. Paul, MN, USA). Product information is listed in Table 1. The overall flow-chart for this experiment is shown in Figure 1, and the group is named as follows: ZMG, zirconia opposing MI Gracefil; DMG, lithium disilicate opposing MI Gracefil; LMG, leucite opposing MI Gracefil; ZGD, zirconia opposing Gradia Direct P; DGD, lithium disilicate opposing Gradia Direct P; LGD, leucite opposing Gradia Direct P; ZEQ, zirconia opposing Estelite Σ Quick; DEQ, lithium disilicate opposing Estelite Σ Quick; LEQ, leucite opposing Estelite Σ Quick; ZFT, zirconia opposing Filtek Supreme Ultra; DFT, lithium disilicate opposing Filtek Supreme Ultra; LFT, Leucite opposing Filtek Supreme Ultra.

To make identical canine-shaped ceramic specimens, a canine-shaped artificial tooth was scanned using a three-dimensional dental scanner (Identica hybrid, Medit, Seoul, Korea) (7 microns accuracy, dual 2.0 MP resolution), and 3D data was generated using CAD software (Exocad Dental CAD; Exocad GmbH, Darmstadt, Germany). All ceramic specimens were fabricated into the canine shape, which had a diameter of 8 mm and a clinical crown length of 15 mm, using a CAD/CAM milling machine (Trione Z, Dio Implants, Pusan, Korea) (Figure 2A), according to the manufacturer’s instructions. Then, post-milling treatment except leucite was conducted (Table 2) and the ceramic specimens were washed using ultrasonic cleaner [34]. Specimens were then washed using an ultrasonic cleaner and embedded into an acrylic resin (Orthodontic resin, Dentsply, Konstanz, Germany) such that 5 mm of specimen cusps were exposed (Figure 2B).

Composite resin specimens were prepared by placing unpolymerized resin pastes into a cuboidal mold (11 × 11 mm by 6 mm in height). Each mold was filled with composite resin in increments of about 2 mm and each layer was light cured for 40 s using a light curing machine (Elipar^TM^ DeepCure-L, 3M ESPE, St. Paul, MN, USA). All composite surfaces were ground and finished with 1000-grit silicon carbide abrasive paper with water cooling. Prepared specimens were fixed using an acrylic resin (Orthodontic resin, Dentsply, Konstanz, Germany) using a uniform mold and the same method used to prepare ceramic specimens to ensure that 2 mm of the heights of composite specimens were exposed (Figure 2C).

Wear tests were conducted using a dual-axis chewing simulator (CS-4.8 masticator, SD Mechatronik, FeldKirchen-Westerham, Germany). In each chamber, composite resin was placed in the upper sample holder and antagonistic ceramic teeth in the lower holder and both were fixed with a fastening screw. A chewing force of 50 N was applied [1,2,35]. According to previous studies, 240,000–250,000 loading cycles are comparable to approximately one-year of chewing [1], and thus, 100,000 chewing cycles corresponds to chewing for 4–5 months. The chewing cycle was carried out in three phases. At first, a 5 kg vertical mass was applied and then the sample was subjected to a horizontal sliding movement of 0.7 mm, and finally allowed to move away from the contact surface. In addition, thermo-mechanical stress was applied by exposing samples to distilled water at 5 °C and at 55 °C for 60 s using a computer-controlled hot/cold water circulation system to simulate the real intra-oral environment [1].

To measure the volumes of ceramic loss, ceramic specimens were scanned with a CAD/CAM three-dimensional dental scanner (Identica hybrid, Medit, Seoul, Korea) before and after wear testing. The three-dimensional data of specimens before and after testing were superimposed such that cusps coincided. stereolithography (STL) files obtained before and after wear testing (Figure 3A,B) were superimposed, and the STL files of wear losses were converted into solid files. Wear volumes were then calculated using the CAD program (Fusion 360, Autodesk, San Rafael, CA, USA) (Figure 3C).

On the other hand, since composite resin specimens were flat, it was not easy to merge STL files before and after wear testing, and thus, wear amounts were determined using weight differences after completely drying specimens. Weight losses of composite resin specimens were measured using an electronic scale (PAG213, Ohaus, Seoul, Korea) accurate to ±10^−4^ grams before and after wear testing.

After completing the three-dimensional evaluation, worn surfaces of composite resin specimens were examined by scanning electron microscope (SEM) (S-3500, Hitachi Ltd., Tokyo, Japan) at 15 keV and magnifications of 35×, 100×, and 1000×.

Means and standard deviations were calculated using statistical software (IBM SPSS Statistics v23; IBM Corp, Armonk, NY, USA). As we performed the test with a small number of specimens, the Kruskal–Wallis test, which is a non-parametric test, was used to evaluate distribution normalities and variance homogeneities, and the significances of differences were determined using the Mann–Whitney U test with Bonferroni correction. Statistical significance was accepted for *p* < 0.05.

## 3. Results

Means and standard deviations of volumetric losses of ceramic tooth samples are shown in Table 2. Figure 4 and Figure 5 show volumetric losses of ceramic teeth after wear testing. Greatest volumetric loss was observed for LEQ (445.8 ± 203.8 mm^3^ × 10^−3^) and lowest composite resin weight loss was observed for ZMG (0.004 ± 0.008 mm^3^ × 10^−3^), but no significant difference was observed between LEQ and ZMG (*p* > 0.05/66 = 0.0008) (Table 3).

When volumetric losses of ceramics were analyzed regardless of composite resin type, monolithic zirconia (0.037 ± 0.838 mm^3^ × 10^−3^) showed significantly less volumetric loss than the lithium disilicate (116.7 ± 214.7 mm^3^ × 10^−3^) or leucite (144.6 ± 208.4 mm^3^ × 10^−3^) (*p* < 0.05/3 = 0.017) (Figure 4A), but no significant difference was observed between lithium disilicate and leucite (*p* > 0.05/3 = 0.017) (Figure 4A). When volumetric losses of ceramics were analyzed according to opposing composite resin type, EQ (289.3 ± 272.8 mm^3^ × 10^−3^) caused significantly more ceramic loss than MG (21.14 ± 54.65 mm^3^ × 10^−3^) or FT (15.34 ± 40.13 mm^3^ × 10^−3^) (*p* < 0.05/6 = 0.0083) (Figure 4B).

MG tended to cause less wear loss of zirconia (0.004 ± 0.008 mm^3^ × 10^−3^) or lithium disilicate (0.008 ± 0.016 mm^3^ × 10^−3^) than of leucite (63.40 ± 84.28 mm^3^ × 10^−3^) (*p* > 0.05/3 = 0.017) (Figure 5A), and GD caused significantly less wear loss of zirconia (0.132 ± 0.134 mm^3^ × 10^−3^) than lithium disilicate (30.00 ± 34.72 mm^3^ × 10^−3^) or leucite (37.40 ± 27.84 mm^3^ × 10^−3^) (*p* < 0.05/3 = 0.017) (Figure 5B). EQ caused significantly less wear loss of zirconia (0.056 ± 0.057 mm^3^ × 10^−3^) than lithium disilicate (422.2 ± 248.3 mm^3^ × 10^−3^) or leucite (445.8 ± 203.8 mm^3^ × 10^−3^) (*p* < 0.05/3 = 0.017) (Figure 5C), and FT caused significantly less wear loss of zirconia (0.000007 ± 0.000014 mm^3^ × 10^−3^) than of or lithium disilicate (14.40 ± 4.278 mm^3^ × 10^−3^) (*p* < 0.05/3 = 0.017) but caused similar wear losses of zirconia and leucite (31.61 ± 70.66 mm^3^ × 10^−3^) (*p* > 0.05/3 = 0.017) (Figure 5D).

The means and standard deviations of the weight losses of dental composite resins after wear tests are presented in Table 3, and the weight losses of different combinations of composite resins and ceramics are shown in Figure 6 and Figure 7. Greatest composite resin weight loss was observed for DEQ (0.94 ± 0.40 mg) and least composite resin losses for ZMG (0.26 ± 0.15 mg) and DGD (0.26 ± 0.15 mg), but no significant difference was observed between DEQ and ZMG, DEQ, and DGD (*p* > 0.05/66 = 0.0008) (Table 3).

When weight losses of composite resins were examined by ceramic type regardless of composite resin type, leucite (0.69 ± 0.49 mg) caused greatest weight loss, followed in decreasing order by zirconia (0.59 ± 0.34 mg) and lithium disilicate (0.52 ± 0.39 mg), but no significant difference was observed between all composite resins (*p* > 0.05/3 = 0.017) (Figure 6A). When weight losses of composite resins were analyzed regardless of ceramic type, EQ (0.95 ± 0.44 mg) showed significantly greater weight losses than FT (0.55 ± 0.24 mg), MG (0.49 ± 0.45 mg), or GD (0.39 ± 0.25 mg) (*p* < 0.05/6 = 0.0083) (Figure 6B).

The weight loss of MG showed the lowest wear value in opposing leucite (0.78 ± 0.58 mg), but there was no statistically significant difference between all opposing ceramics (*p* > 0.05/3 = 0.017) (Figure 7A). The weight loss of GD showed the lowest wear value in opposing zirconia (0.54 ± 0.29 mg), but there was no statistically significant difference between all opposing ceramics (*p* > 0.05/3 = 0.017) (Figure 7B). The weight loss of EQ showed the lowest wear value in opposing leucite (1.07 ± 0.53 mg), but there was no statistically significant difference between all opposing ceramics (*p* > 0.05/3 = 0.017) (Figure 7C). The weight loss of FT showed the lowest wear value in opposing zirconia (0.72 ± 0.16 mg), but there was no statistically significant difference between all opposing ceramics (*p* > 0.05/3 = 0.017) (Figure 7D).

SEM images of the composite resin after wear test are shown in Figure 8. Composite resins opposed to zirconia showed minor pitting, but generally smooth surfaces (ZMG, ZGD, ZEQ, and ZFT). Those exposed to lithium disilicate showed step-like fractures in the sliding direction (DMG, DGD, DEQ, and DFT), whereas those opposing leucite showed large surface pits, many microcracks, and crater-like defects in the sliding direction (LMG, LGD, LEQ, and LFT). In addition, the wear patterns of EQ were rougher in the sliding direction than those of other composites and were obvious even at low magnification (35×).

## 4. Discussion

Dental wear is a complex process that is influenced by numerous factors and depends on physiological and pathological mechanisms [2,36]. Resistance to occlusal wear is an important consideration for the clinical success of oral prosthetic restorations. Restorative material wear should match as closely as possible that of natural enamel [2,25]. This is because wear of restorative materials causes abnormal loading on occlusal surface and possibly loss of occlusal vertical dimensions, which can lead to problems, such as temporomandibular joint disorders, masticatory muscle fatigue, changes in mandibular movement path, and esthetic problems [1,37]. In addition, a restorative material should not cause abrasion of opposing natural teeth or restorative materials [38]. A variety of tests—such as the two-body wear [36], three-body wear [15,39], and rotating sliding wear tests [17] and toothbrush simulation [15]—have been used to evaluate the wear properties of restorative materials. However, these tests cannot accurately simulate wear in the oral cavity [37,40]. However, in the present study, the two-body wear test used was used because it is practical, cost-effective and widely used [34,37], and provides a means of accessing friction and fatigue wear due to direct contact between teeth during swallowing or non-chewing movements [41]. Clinically, although most restorative material or tooth loss may be due to direct contact between teeth, between teeth and restorative materials, or between restorative materials, these losses can also be caused by other factors, such as abrasion or erosion [42]. An applying load of 50 N can be considered representative of physiological occlusal forces in a normal individual, but not in bruxism patients [1,2,35]. Furthermore, during wear testing, it is also essential to continue to exchange water to remove particles that are worn away from the specimen it is important to remove particles produced by wear by continuously exchanging test water [43].

In this study, ceramic specimens were prepared by CAD/CAM using maxillary canine-shaped artificial teeth. Composite resin specimens were produced from direct restorative composite resins. With regard to the increasing demand for the use of esthetic tooth restorative materials, the experiment was designed to mimic clinical situations in which composite resin restorations are opposed by ceramic restorations in the oral cavity.

Wear of ceramic specimens was analyzed by measuring volume loss. The ceramic specimens used were milled using a CAD/CAM method to produce identically shaped artificial crowns and cusps. In a previous study on ceramic wear against enamel, it was considered that standardization of antagonistic enamel morphology would not have a significant effect on experimental results based on the assumption that amount of opposing enamel wear is independent of enamel morphology [43]. Nonetheless, specimen volume losses are easily accessed by subtracting specimen volumes after wear testing from pre-test values [34]. Maximum height loss is more clinically relevant because the interocclusal distance between maxillary and the mandibular teeth is determined by occlusal contact points [44]. In previous studies, three-dimensional laser, mechanical, optical, and flat-sample-based methods were used to quantify the wear of dental materials [45]. However, it has been shown that maximum height loss and volume loss are well correlated, and thus, volume loss measurements provide an useful means of evaluating material wear properties [43,46].

In the present study, composite resin wear loss was evaluated by measuring specimen weight losses, which is a technique that has been used previously to determine the wear losses of restorative materials [34]. Initially, it was our intention to determine volumetric losses of composite resin specimens after wear testing as described for ceramic losses. However, since the composite resin specimens had a cuboid shape, it was difficult to accurately superimpose three-dimensional pre- and post-test files at the same positions.

We found ceramic volumetric wears were not always significantly difference, but zirconia was found to wear less than lithium disilicate and leucite, which concurs with the results of several previous studies on ceramic wear when opposed to enamel [43,47,48,49]. However, in terms of composite resin wear, no significant difference or differential tendency was observed for the three ceramics.

The wear mechanisms of composite resins and ceramics depend on material types. Ceramic wear is induced by microfracture, whereas composite resin wear is caused by adhesion and plastic deformation [10,49]. In the 1980s, it was thought that ceramic surface hardness was correlated with opposing natural enamel wear [50], but this notion has been recently been disproven [25]. Ceramic surface roughness is also considered to be an influential factor, and in the context of enamel wear opposing zirconia or glass ceramics, it has been reported that zirconia is much less susceptible to surface microfracture [51].

Wear of a composite resin is affected by filler content, the bond between the filler and the matrix, and bonding of the coupling agent. Various filler systems have been developed to improve the mechanical properties of composites, and in recent years, composite resins containing nano-sized fillers have been developed to increase gloss and mechanical properties [52]. In the present study, composite resins showed material-dependent wear loss. In particular, EQ exhibited higher wear losses than the other composite resins but not for all types of opposing ceramics. It is generally known that larger filler particle size result in more filler protrusion from restoration surfaces subjected to wear, and that these particles are likely to be scuffed off by friction forces [10,52]. However, filler sizes of the composite resins used in the present study were in the range 100 to 850 nm, which is considered small, and thus, they were supposed to produce smoother surfaces and to pack more closely, and thus, to better resist friction forces. In SEM images after wear testing, some traces of sliding movement were observed along the migration path of the opposing ceramic tip for all composite resins. However, neither large filler particle detachment or matrix fractures were observed on EQ and FT, which had mean filler sizes of 200 and 100 nm, respectively. On the other hand, voids and interfacial defects were observed in SEM images of MG and GD with larger filler particle sizes due to fracture of the particles such as filler and prepolymerized filler. Furthermore, GD and EQ are hybrid resins and contain prepolymerized fillers, and it was suggested in a previous study that such composite resins may be worn due to displacement of matrix constituents rather than actual substance loss [52]. Prepolymerized fillers are organic or inorganic hybrids, which are incorporated into resins to reduce polymerization shrinkage. They are also surface-treated with silane, which promotes matrix bonding and reduces stress cracking and surface filler losses [52].

Flexural strength is related to fracture-related properties, such as fracture resistance and material elasticity, and has been proposed to also offer a means of evaluating brittleness. Fillers with smaller particles at higher filler loadings are likely to have higher flexural strengths [52]. If the amount of energy required to fracture a material is relatively small, cracks will occur more easily and wear will occur more quickly. Ferracane [53] suggested that flexural strength and fracture toughness are associated with fatigue fracture, because cumulative repeated loading causes material failure due to damage accumulation. In the present study, EQ had a lower flexural strength than the other composite resins examined, which suggests that the resin fractured at lower flexural stress.

SEM of composite resin specimens after abrasion testing provide clues of the wear mechanism. Wear of composite resin materials is generally described by two processes, that is, abrasive and fatigue wear [19,38]. Abrasive wear describes the loss of a material’s surface due to friction between it and the surface of another material [38,54]. Fatigue wear, on the other hand, is the result of subsurface damage and the destruction of intermolecular bonds [54,55]. When a material is subjected to a force beyond its shear or tensile strength, micro-cracks appear below its surface, and these eventually result in fracture and the loss of material [49,55]. In particular, fatigue wear occurs at bonds between matrices and fillers and results in the loss of filler particles.

The wear mechanism of the composite resin can be estimated by observing the wear track on the SEM image after the wear test. In the present study, EQ showed most material wear loss and exhibited a distinct parallel wear pattern along the contact directions. These distinct parallel rhombic patterns are more pronounced when confronted with leucite. Cracks visible on composite resin surfaces are created when material is dragged across their surfaces [54]. Lawson et al. [51] found that as filler concentrations in composite resins increase beyond specific thresholds, the wear mechanism switches from fatigue to wear.

## 5. Conclusions

Within the limitations of this in vitro study, zirconia showed less volumetric loss than lithium disilicate or leucite when opposed to composite resins. Although some of the composite resins used in this study showed considerable wear, no direct association was detected between the amount of composite resin wear and filler properties or composite resin mechanical properties. This implies that the size and content of different matrices and fillers are very various, and it is extremely difficult to directly relate the effect of this to the wear properties of the composite resins. On the other hand, lithium disilicate and leucite have a lower wear resistance than zirconia, so additional caution is needed. Since wear properties appear to be dependent on ceramic and opposing composite resin types, it is evident that the choice of appropriate restorative materials needs to be made based on consideration of clinical situations.

## Figures and Tables

**Figure 1 materials-12-01839-f001:**
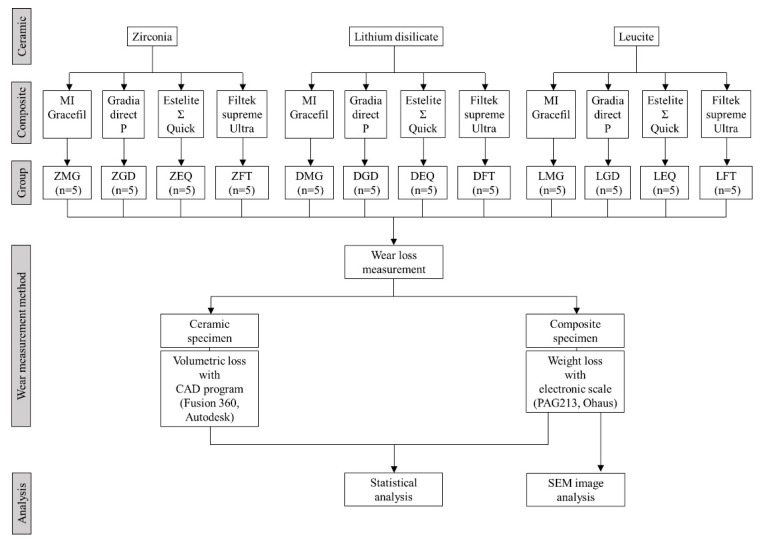
Flow-chart of the study.

**Figure 2 materials-12-01839-f002:**
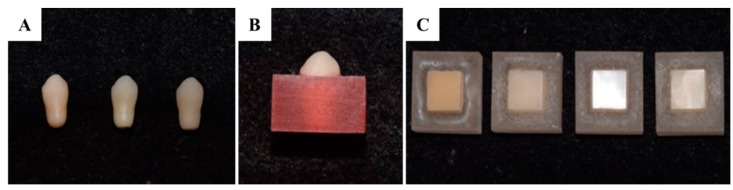
(**A**) Preparation of the dental ceramic specimen (from left, monolithic zirconia; lithium disilicate; leucite). (**B**) Dental ceramic specimen embedded in autopolymerizing acrylic resin mold. (**C**) Preparation of the dental composite specimen (from left, MG, MI Gracefil; GD, Gradia Direct P; EQ, Estelite Σ Quick; FT, Filtek Supreme Ultra).

**Figure 3 materials-12-01839-f003:**
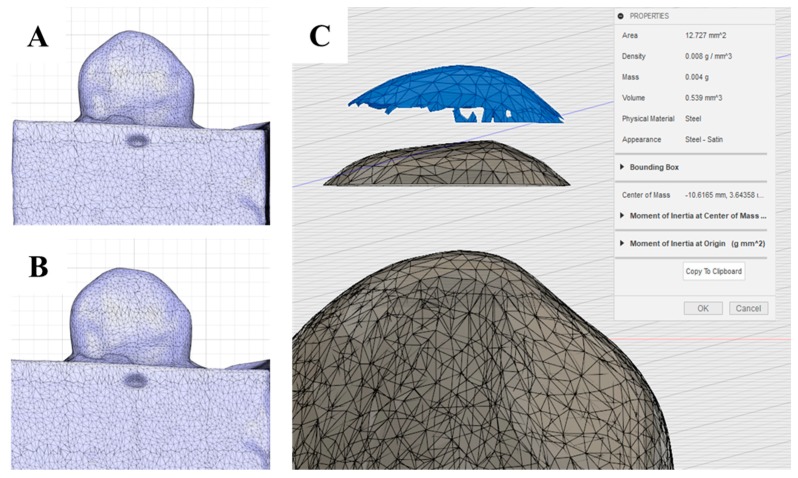
Measuring volumetric loss of ceramic specimens using three-dimensional images: (**A**) STL file of the ceramic specimen before wear testing. (**B**) STL file after wear testing. (**C**) Wear volume measurement; wear volume (blue part; top) = STL file before wear (bottom)-STL file after wear (middle).

**Figure 4 materials-12-01839-f004:**
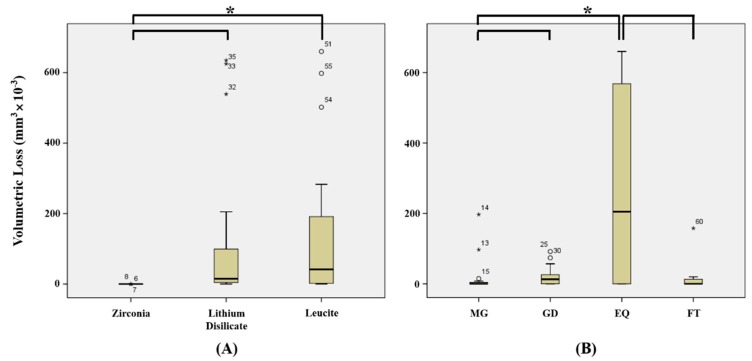
Volumetric losses of ceramic specimens (mm^3^ × 10^−3^). (**A**) Mean volumetric loss of ceramic regardless of composite resin type. (**B**) Mean volumetric loss of ceramic according to composite resin type. MG, MI Gracefil; GD, Gradia Direct P; EQ, Estelite Σ Quick; FT, Filtek Supreme Ultra. * Statistical significance.

**Figure 5 materials-12-01839-f005:**
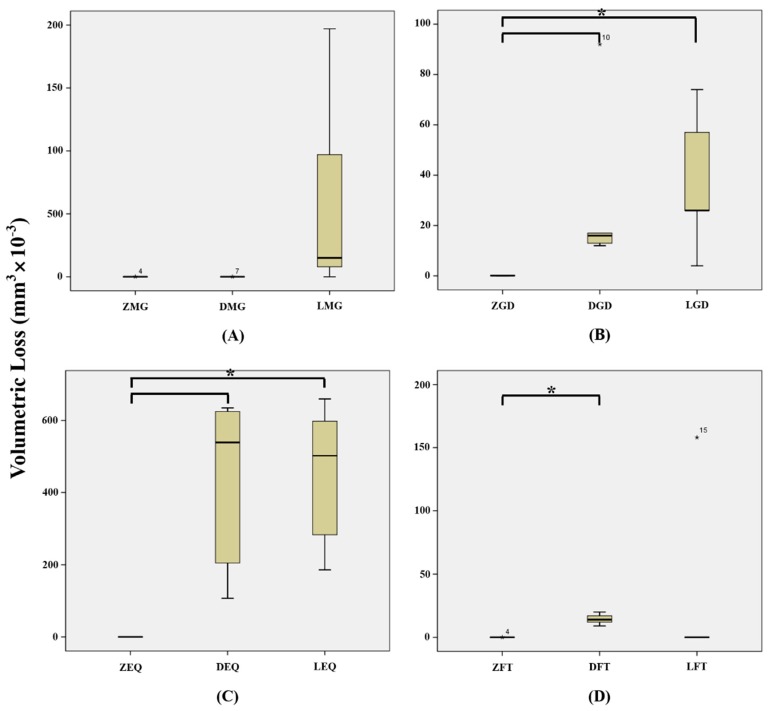
Volumetric losses of ceramic specimens (mm^3^ × 10^-3^). (**A**) Volumetric losses of ceramics when opposing to MG. (**B**) Volumetric losses of ceramics when opposing to GD. (**C**) Volumetric losses of ceramics when opposing to EQ. (**D**) Volumetric losses of ceramics when opposing to FT. ZMG, zirconia opposing MI Gracefil; DMG, lithium disilicate opposing MI Gracefil; LMG, leucite opposing MI Gracefil; ZGD, zirconia opposing Gradia Direct P; DGD, lithium disilicate opposing Gradia Direct P; LGD, leucite opposing Gradia Direct P; ZEQ, zirconia opposing Estelite Σ Quick; DEQ, lithium disilicate opposing Estelite Σ Quick; LEQ, leucite opposing Estelite Σ Quick; ZFT, zirconia opposing Filtek Supreme Ultra; DFT, lithium disilicate opposing Filtek Supreme Ultra; LFT, leucite opposing Filtek Supreme Ultra. * Statistical significance.

**Figure 6 materials-12-01839-f006:**
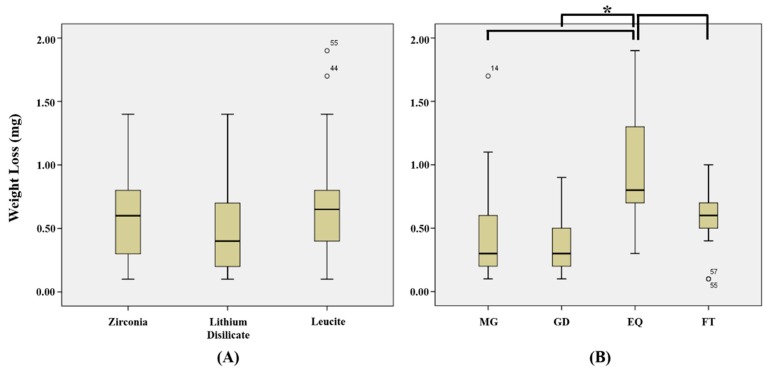
Weight losses of composite resin specimens (mg): (**A**) Mean weight loss of composite resin according to ceramic type. (**B**) Mean weight loss of composite resin regardless of ceramic type. MG, MI Gracefil; GD, Gradia Direct P; EQ, Estelite Σ Quick; FT, Filtek Supreme Ultra. * Statistical significance.

**Figure 7 materials-12-01839-f007:**
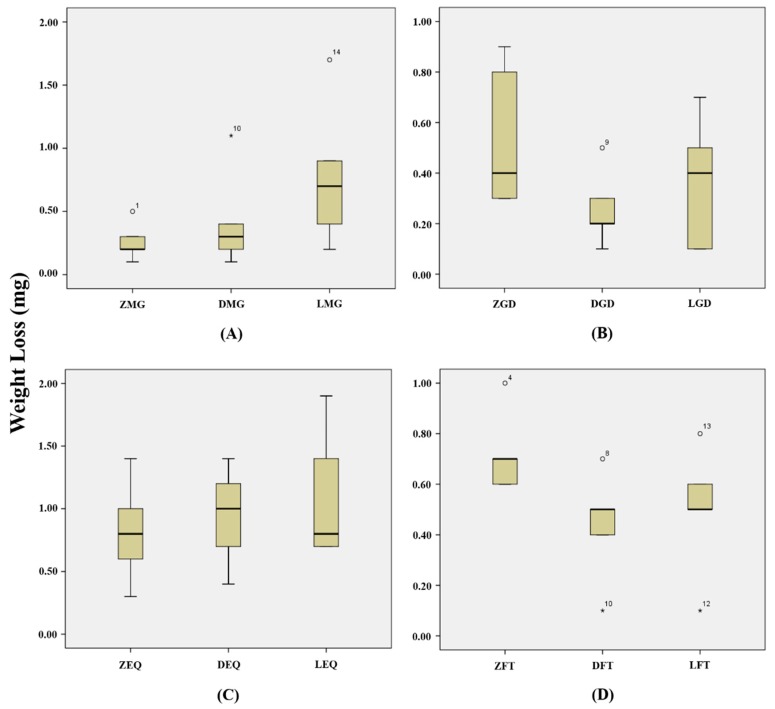
Weight losses of composite resin specimen (mg): (**A**) Weight losses of MG when opposing to ceramics; (**B**) Weight losses of GD when opposing to ceramics; (**C**) Weight losses of EQ when opposing to ceramics; (**D**) Weight losses of FT when opposing to ceramics. ZMG, zirconia opposing MI Gracefil; DMG, lithium disilicate opposing MI Gracefil; LMG, leucite opposing MI Gracefil; ZGD, zirconia opposing Gradia Direct P; DGD, lithium disilicate opposing Gradia Direct P; LGD, leucite opposing Gradia Direct P; ZEQ, zirconia opposing Estelite Σ Quick; DEQ, lithium disilicate opposing Estelite Σ Quick; LEQ, leucite opposing Estelite Σ Quick; ZFT, zirconia opposing Filtek Supreme Ultra; DFT, lithium disilicate opposing Filtek Supreme Ultra; LFT, leucite opposing Filtek Supreme Ultra.

**Figure 8 materials-12-01839-f008:**
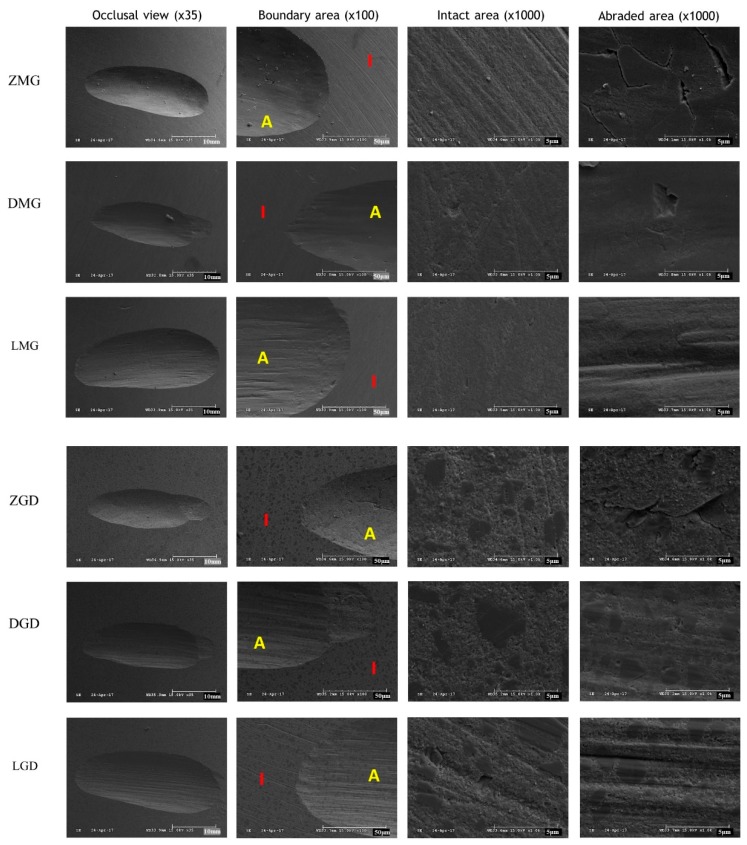
Representative SEM images of composite resin surfaces: ZMG, zirconia opposing MI Gracefil; DMG, lithium disilicate opposing MI Gracefil; LMG, leucite opposing MI Gracefil; ZGD, zirconia opposing Gradia Direct P; DGD, lithium disilicate opposing Gradia Direct P; LGD, leucite opposing Gradia Direct P; ZEQ, zirconia opposing Estelite Σ Quick; DEQ, lithium disilicate opposing Estelite Σ Quick; LEQ, leucite opposing Estelite Σ Quick; ZFT, zirconia opposing Filtek Supreme Ultra; DFT, lithium disilicate opposing Filtek Supreme Ultra; LFT, leucite opposing Filtek Supreme Ultra. I, Intact area; A, Abraded area.

**Table 1 materials-12-01839-t001:** List of materials used and their characteristics.

**Ceramic Specimens**						
**Product**	**Composition**		**Biaxial Flexural Strength (MPa)**	**Fracture Toughness (MPa·m^1/2^)**	**Vickers Hardness (GPa)**	**Manufacturer**
Zirtooth^TM^ Fulluster (Monolithic Zirconia)	Zirconium dioxide, hafnium dioxide, yttrium trioxide, pigment, organic binder		1200	5.5	13	HASS Corp., Gangwon-do, Korea
Rosetta^®^ SM (Lithium Disilicate)	Lithium oxide, silicon dioxide, boron trioxide, phosphorus pentoxide, pigment, organic binder		440	2.25	5.8	HASS Corp., Gangwon-do, Korea
Rosetta^®^ BM (Leucite)	Potassium oxide, silicon dioxide, boron trioxide, titanium dioxide, aluminum oxide		100	1.42	6	HASS Corp., Gangwon-do, Korea
**Composite Resin Specimens**						
**Product**	**Filler**	**Matrix**	**Filler Content (wt %)**	**Filler Size (nm)**	**Flexural Strength (MPa)**	**Manufacturer**
MI Gracefil (Nano-Hybrid Filler Resin)	Barium silica glass	Bis-MEPP, UDMA	82	300	170	GC, Tokyo, Japan
Gradia Direct P (Micro-Hybrid Filler Resin)	Barium silica glass, silica dioxide, prepolymerized filler	UDMA	76	850	129	GC, Tokyo, Japan
Estelite Σ Quick (Micro-Hybrid Filler Resin)	Silica-zirconia supra-nano monodispersing spherical, PFSC	Bis-GMA, TEGDMA, UDMA	78	200	110	Tokuyama Dental, Tokyo, Japan
Filtek Supreme Ultra (Nano-Filler Resin)	Zirconium/silica non-agglomerated particles	Bis-GMA, UDMA, Bis-EMA, TEGDMA	78	100	161	3M ESPE, MN, USA

All information was supplied by the manufacturers. Bis-EMA, bisphenol A-lolyethylene diethe dimethacrylate; Bis-GMA, bisphenol A-glycidyl methacrylate; Bis-MEPP, bisphenol-A-ethoxylate dimethacrylate; PFSC, prepolymerized filler of silica composite; TEGDMA, triethyleneglycol dimethacrylate; UDMA, urethane dimethacrylate.

**Table 2 materials-12-01839-t002:** Post milling procedure of the lithium disilicate and zirconia.

Group	Start Temp (°C)	Heat Rate (°C/min)	High Temp (°C)	Hold (min)	Vacuum (°C)	Vacuum off (°C)
Lithium disilicate	400	60	840	10	550	845
Zirconia	1450	4	1550	120	-	-

**Table 3 materials-12-01839-t003:** Means ± standard deviations (SD) of volumetric losses of dental ceramic teeth and weight losses of dental composite resins after wear test.

Group	Mean ± SD	
	Ceramic Volumetric Loss (mm^3^ × 10^−3^)	Composite Resin Weight Loss (mg)
ZMG	0.004 ± 0.008	0.26 ± 0.15
ZGD	0.134 ± 0.134	0.54 ± 0.29
ZEQ	0.006 ± 0.006	0.82 ± 0.41
ZFT	0.013 ± 0.014	0.72 ± 0.16
DMG	0.008 ± 0.016	0.42 ± 0.40
DGD	30.00 ± 34.72	0.26 ± 0.15
DEQ	422.2 ± 248.2	0.94 ± 0.40
DFT	14.40 ± 4.278	0.44 ± 0.22
LMG	63.40 ± 84.28	0.78 ± 0.58
LGD	37.40 ± 27.84	0.36 ± 0.26
LEQ	445.8 ± 203.8	1.1 ± 0.53
LFT	31.61 ± 70.66	0.50 ± 0.25
*p*	<0.001	0.006

*p* value result from Kruskal–Wallis test at the 5% significance level (α = 0.05). Multiple comparisons used Mann–Whitney U test with Bonferroni correction (α = 0.05/66 = 0.0008) and no statistical significance was observed between the groups. ZMG, zirconia opposing MI Gracefil; DMG, lithium disilicate opposing MI Gracefil; LMG, leucite opposing MI Gracefil; ZGD, zirconia opposing Gradia Direct P; DGD, lithium disilicate opposing Gradia Direct P; LGD, leucite opposing Gradia Direct P; ZEQ, zirconia opposing Estelite Σ Quick; DEQ, lithium disilicate opposing Estelite Σ Quick; LEQ, leucite opposing Estelite Σ Quick; ZFT, zirconia opposing Filtek supreme Ultra; DFT, lithium disilicate opposing Filtek Supreme Ultra; LFT, leucite opposing Filtek Supreme Ultra.

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
