# Peer review of "Wear Characteristics of Dental Ceramic CAD/CAM Materials Opposing Various Dental Composite Resins"

_materials, 2019, doi:10.3390/ma12111839_

Round 1

Reviewer 1 Report

Dear Authors, below are my comments about the submitted manuscript

1. I would indicate in the title or in the abstract that ceramic materials used in the study were obtained through CAD/CAD procedures.

2. Pag. 1 line 42: the statement “structural stability” regarding zirconia is way too bold, since its well known ageing process. I suggest eliminating that sentence or thoroughly modify it.

3. Pag. 2 line 50: maybe you mean “them” instead of “then”?

4. Pag. 2 line 59: I do not get the mean of the sentence “and that surface abrasion caused by zirconia can greatly reduce the wear of opposing enamel”. Please clarify.

5. At the end of the Introduction section, the null hypothesis is missing. Its presence is mandatory to carry out an inferential statistical analysis. Please add it.

6. Pag. 4 line 97: UDMA and UEDMA have the same description.

7. Why do you choose to shape the ceramic as canines and the resin composite as cuboidal masses to perform the experiment? Did you base on previous study or was it your own idea? In both cases, please clarify.

8. The description of the post-milling treatment of the three group of ceramic is missing.

9. You have to report the accuracy, precision and resolution of the CAD/CAM 3D dental scanner used to measure the volume of ceramic loss for a better interpretation of the results.

10. I do not get the choice to use the Kruskal-Wallis test to evaluate distribution normalities. You need to assess the distribution type previously by a normality distribution test and then perform the parametric or non parametric specific test. Otherwise, you can use a non parametric test when the sample size is too small to know if a normal distribution of the data exists. Please clarify. Moreover, the Kruskal-Wallis test is used to compare the MEDIAN of different groups. Please furnish a comprehensive explanation of how you managed the statistical analysis.

11. Pag. 13 figure 8 should be modified. The images need to be enlarged for a better appreciation of details of each image.

12. I cannot appreciate a clear, definite, sharp conclusion. Can you please provide it, maybe re-setting up the Conclusion section?

Author Response

1. I would indicate in the title or in the abstract that ceramic materials used in the study were obtained through CAD/CAD procedures.
Answer) Thank you for your sincere advice. According to your comment, I changed the title like below. And I also changed the word used in the abstract correspondingly.

Wear characteristics of dental ceramic CAD/CAM materials opposing various dental composite resins

2. Pag. 1 line 42: the statement “structural stability” regarding zirconia is way too bold, since its well known ageing process. I suggest eliminating that sentence or thoroughly modify it.
Answer) Thank you for your advice. I modified the description as follows.

In addition, these materials excellent physical properties and thus, are widely used in fixed prostheses.

3. Pag. 2 line 50: maybe you mean “them” instead of “then”?
Answer) Thank you for the mention. I misspelled it and corrected the mistake.

4. Pag. 2 line 59: I do not get the mean of the sentence “and that surface abrasion caused by zirconia can greatly reduce the wear of opposing enamel”. Please clarify.
Answer) Thank you for your comment. I changed the description like below.

and the process for polishing the surface of zirconia can further reduce the wear of opposing enamel.

5. At the end of the Introduction section, the null hypothesis is missing. Its presence is mandatory to carry out an inferential statistical analysis. Please add it.
Answer) Thank you for your mention. I added the null hypothesis as below.

The null hypothesis tested for this study was that no significant difference could be detected in the wear properties among the materials under this study.

6. Pag. 4 line 97: UDMA and UEDMA have the same description.
Answer) I appreciate your comment. Both UDMA and UEDMA are synonyms, meaning urethane dimethacrylate, and I modified them by UDMA.

7. Why do you choose to shape the ceramic as canines and the resin composite as cuboidal masses to perform the experiment? Did you base on previous study or was it your own idea? In both cases, please clarify.
Answer) I appreciate your comment. As I mentioned from Pag. 14 line 288, according to a previous study, the standardization of antagonistic enamel morphology would not affect the experimental results, and I thought identification of the shape of the ceramic specimens would simplify the measurement of wear loss during the study. And for the resin specimen, I designed the shape of the specimen based on the previous study (reference number 40).

8. The description of the post-milling treatment of the three group of ceramic is missing.
Answer) Thank you for your valuable comment. I added post-milling treatment information in table 2 and additionally described.

9. You have to report the accuracy, precision and resolution of the CAD/CAM 3D dental scanner used to measure the volume of ceramic loss for a better interpretation of the results.
Answer) Thank you for your valuable comment. I updated the resolution and accuracy information of the 3D scanner I used.

10. I do not get the choice to use the Kruskal-Wallis test to evaluate distribution normalities. You need to assess the distribution type previously by a normality distribution test and then perform the parametric or non parametric specific test. Otherwise, you can use a non parametric test when the sample size is too small to know if a normal distribution of the data exists. Please clarify. Moreover, the Kruskal-Wallis test is used to compare the MEDIAN of different groups. Please furnish a comprehensive explanation of how you managed the statistical analysis.
Answer) When the Kruskal-Wallis test was performed on the 12 groups in table 2, the p values of each tests were obtained as shown in table 2, and there was no statistical significance when using the Mann-Whitney U test for a post-test. As you said, I had to perform a non-parametric test because it didn’t have the normal distribution with only a small number of specimens, and I took the Kruskal-Wallis test for a non-parametric test. Based on your opinion, I modified the description.

11. Pag. 13 figure 8 should be modified. The images need to be enlarged for a better appreciation of details of each image.
Answer) Thank you for your comment. I changed the images to be enlarged one.

12. I cannot appreciate a clear, definite, sharp conclusion. Can you please provide it, maybe re-setting up the Conclusion section?
Answer) I appreciate your helpful advice. I rewrite the conclusion part.

Reviewer 2 Report

 My recommendation is to be published after revision of the  manuscript according to folloving :

1.The title starting with the term”in vitro” it is not the   best choice for such investigation with CAD/CAM preparation from maxillary canine-shaped  artificial teeth.

2.In the table 1 with the list  of materials used and their characteristics will be better to mention exactly the provider of characteristics  with details

 3.It is a need to complete flow-chart of the study from figure 1 with examination worn surfaces of composite resin specimens by scanning electron microscope (SEM) and statistical analysis which are indeed important parts of this paper

4.It  will be in the benefit of the paper to present SEM investigation  with more quantified data,

Author Response

1.The title starting with the term”in vitro” it is not the   best choice for such investigation with CAD/CAM preparation from maxillary canine-shaped  artificial teeth.
Answer) Thank you for your sincere advice. According to your comment, I changed the title like below.

Wear characteristics of dental ceramic CAD/CAM materials opposing various dental composite resins

2.In the table 1 with the list  of materials used and their characteristics will be better to mention exactly the provider of characteristics  with details
Answer) Thank you for your kindness. All information was supplied by the manufacturers. I added the provider information in the article.

3.It is a need to complete flow-chart of the study from figure 1 with examination worn surfaces of composite resin specimens by scanning electron microscope (SEM) and statistical analysis which are indeed important parts of this paper
Answer) Thank you for your advice. I modified the flow chart.

4.It  will be in the benefit of the paper to present SEM investigation  with more quantified data,
Answer) I appreciate your sincere advice. I attempted a quantitative analysis of the amount of wear loss, but it was very difficult to obtain the reliability of the quantitative analysis results because it was hard to have reliable reference point.

Round 2

Reviewer 1 Report

Dear Authors,
thanks for the modifications